# The Southern Hemisphere Blocking Index Revisited

**Adrián E. Yuchechen** [1,*], **S. Gabriela Lakkis** [2] **and Pablo O. Canziani** [1]

[1] Unidad de Investigación y Desarrollo de las Ingenierías (UIDI), Consejo Nacional de Investigaciones Científicas y Técnicas (CONICET), Facultad Regional Buenos Aires (FRBA), Universidad Tecnológica Nacional (UTN), Mozart 2300, Ciudad Autónoma de Buenos Aires C1407IVT, Argentina

[2] Facultad de Ingeniería y Ciencias Agrarias, Pontificia Universidad Católica Argentina, UTN, FRBA, UIDI, Av. Alicia Moreau de Justo 1500, Ciudad Autónoma de Buenos Aires C1107AFD, Argentina

[*] Correspondence: aeyuchechen@frba.utn.edu.ar; Tel.: +54-11-4867-7500 (ext. 7268)

**Abstract:** An updated climatology for the occurrence of blockings in the Southern Hemisphere (SH) using daily reanalysis data from 1948–2021 is presented. Geopotential height ($z$) at 500 hPa was the working variable. The blocking index (BI) was defined for every 2.5° of longitude whenever $z(35° \text{ S}) - z(50° \text{ S}) < 0$. The results were organized in longitudinal bands of a width of 10° in order to compare them with previous findings. The primary region for the occurrence of blockings was located around the date line, with a secondary region in southern South America (SSA) and its vicinities, with a third-rank region situated in southern Africa and its surroundings. The results were also stratified by the intensity and duration (persistence) of the events, and the annual and seasonal differences were discussed. Additionally, three different areas were defined to study the distribution of the blockings therein, with the Pacific region (110° E–80° W) having the maximum intensities and longest durations. Linear trends were estimated for the annual and the seasonal time series of the BI and for the number of episodes. On an annual basis, more frequent and the strongest events are expected at 180° E and their surroundings in the future. An alternative BI, using radiosonde data, was built for SSA at 58.50° W. The time evolution of this index was in general agreement with the one estimated from the reanalysis data at some longitudes.

**Keywords:** blocking; blocking index; reanalysis data; radiosonde data; Southern Hemisphere; South America

## 1. Introduction

Blocking anticyclones, or blockings for short, are synoptic-scale-stationary and persistently warm high-pressure systems that obstruct the usual passage of other eastward-moving weather systems. Blockings are favored during winter over the oceans. They are located within the westerly wind belt, downstream of the principal storm tracks, where the flow in the middle troposphere is climatologically diffluent. The interactions of small-scale systems with the diffluent flow enhance the diffluence, eventually leading to the establishment of an aloft anticyclonic circulation [1]. Blockings have an equivalent barotropic structure with a closed anticyclone at lower levels and a ridge in the upper troposphere [2] (p. 309).

Traditionally linked to fair weather [1,3], droughts and heatwaves [1,4–8], and pollution trapping [1,3,9], blockings are also frequently associated to areas of cyclogenesis and cut-off lows [10,11]. Actually, the dipole pattern in which a low-pressure system locates equatorward of a high-latitude blocking anticyclone [12] (p. 605) is one of the three blocking types. In this structure, which is also termed the "high-over-low" block in the Northern Hemisphere (NH)—the Southern Hemisphere (SH) counterpart should be termed "low-over-high"—there is a splitting of the westerly current [13]. The other two blocking types are the zonally-oriented omega blocking, in which a high-latitude anticyclone is flanked by two lower-latitude cyclones, one to the east and the other to the west, and the stationary high-amplitude ridge [13].

The criteria for the identification of blockings exploit their stationary characteristics, as, in these events, the zonal movement of short waves is halted [13] (p. 79), along with their aforementioned spatial structure, or both. Although being an arbitrary constraint, stationary anticyclones are considered blockings if they persist for five days or more [2] (p. 309). The detection of zonal index-type blockings is based on the assessment of the reversal of one- or two-dimensional height gradients at a certain pressure level, typically 500 hPa [14,15]. Another approach is the thresholding methodology, whereby blocking events are defined by exceeding critical pre-specified values (e.g., [16,17], and [18], and references therein). Following the thinking of general circulation, in terms of the potential temperature $(\theta)$/potential vorticity tandem, which was introduced in the early 1990s [19], blockings have also been defined in the literature by means of the reversal in the meridional gradient of $\theta$ over the dynamical tropopause [20].

After one of the earliest research efforts, which studied blockings in the NH mid-latitude middle troposphere at 500 hPa [21], this pressure level has traditionally been used by the scientific community in order to identify blockings, both in the NH [22–30] and in the SH [14,27,31–34]. When compared to the NH, in the SH, the anomalous flows, including blockings, are of lower amplitude and lesser persistence ([35] and references therein). This is partly attributed to stronger westerlies at mid to high latitudes in the SH [36]. The primary region for blocking activity in the SH owes its existence to the split of the upper-level westerlies and locates in the eastern Pacific Ocean [1,14,35,36]. Other maxima occur southeast of South America (SA) and Africa [35,37].

This paper presents an updated climatology of the SH for the blocking index (BI) defined in [14] within the 74-year period between 1948 and 2021 using reanalysis data. Additionally, given that southeastern SA is a region prone to blockings across the year, we took advantage of the location of the two upper-air stations close to 60° W in the area, with the latitudes of each station approximately in coincidence with the ones used in the definition of the foregone BI, in order to construct an alternative BI from radiosonde data. The BI and the radiosonde-derived BI (RBI) were confronted with each other, and the discrepancies between these two indices were assessed.

## 2. Materials and Methods

The 500 hPa geopotential height data used in this paper encompass the 74-year period from 1948–2021. This dataset was obtained from the NCEP/NCAR Reanalysis 1 project [38] through its webpage [39]. Each daily mean archive has global coverage and is arranged in a 73 × 144 (latitude × longitude) grid. The data spans the 90° N–90° S latitude and 0° E–357.5° E longitude ranges, with 2.5° steps in both cases. The number of daily archives between 1 January 1948 and 31 December 2021 is 27,010.

Following [14], the 144 longitudinal values were Fourier-analyzed on a daily basis at the 29 available latitudes between 20° S and 90° S. Following this, the values of the geopotential height, $z$, at each of the foregoing latitudes can be expanded without error in the Fourier series:

$$z_i(\varphi) = [z(\varphi)] + \sum_{n=1}^{72} [a_n \cos(k_n i) + b_n \sin(k_n i)] \quad (i = 1, \ldots, 144) \tag{1}$$

In the above equation $i$ denotes the longitude ($i = 1$ and $i = 144$ correspond to 0° E and 357.5° E, respectively), $\varphi$ is the latitude, $[z(\varphi)]$ indicates the latitudinal mean, $k_n = 2\pi n/\lambda$ is the wavenumber associated to the zonal wavelength $\lambda/n$ (where $\lambda$ is the fundamental wavelength that spans an entire circle of latitude), and $a_n$ and $b_n$ are the expansion coefficients. The values of $z$ are expressed in meters. From now on, the wavelength associated to the $n-$th term in (1) will be referred to as wave $n$. The variance represented by the $n-$th term of the summation in (1) is proportional to $a_n^2 + b_n^2$ [40] (p. 69).

Shown in Figure 1 is the 74-year averaged contribution to the total variance of the first 10 waves at two different latitudes that define de BI below, namely 35° S and 50° S, for summer and for winter. The seasons are defined relative to the SH with the quarters

December-January-February (DJF), March-April-May (MAM), June-July-August (JJA) and September-October-November (SON) standing for summer, autumn, winter and spring, respectively. In DJF (Figure 1a) the greatest variances at 35° S were represented by waves 4 and 5, slightly below 16% each, closely followed by wave 1 (below 15%). It can also be seen that in general, the variance decreased with the increasing value of $n$ (i.e., shorter waves). At 50° S, there was a dominance of wave 1, having 27% of the variance, whereas waves 3 and 4 altogether took on another 33%, and there was also a decrease in the variance for shorter waves. As for JJA (Figure 1b), wave 1 dominated at both latitudes. In the case of 35° S, the variance for $n \geq 2$ showed a mild decrease in favor of $n = 1$ when compared to DJF. The opposite happened at 50° S, where something related to a greater variance during the season, owing to an equatorward advance of baroclinicity which is mostly connected to stronger jets [41,42], i.e., shorter waves, was more predominant during the season [43]. In particular, wave 3 played a key role in the majority of the blocking events [36], something that was closely related to land-sea contrasts in the heating of the lower levels of the atmosphere [44]. The dominance of waves 1 and 3, both in DJF and in JJA, was also important, as in many situations, their interplay is responsible for the establishment of persistent anomalies in the SH ([44] and references therein).

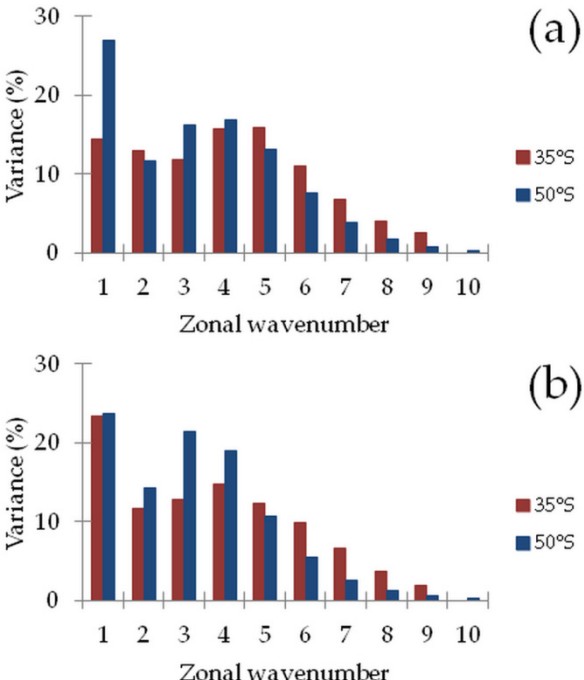

**Figure 1.** Average variance of waves 1 to 10 at 35° S and 50° S during two different seasons within the 74-year period: (**a**) DJF; (**b**) JJA.

Given that blockings span extratropical areas ([45] and references therein) the average picture depicted in Figure 1 can be broken down in order to show the annual contribution of each wave in the 20° S–90° S range. Figure 2 shows seasonal Hövmoller diagrams with the contribution of wave 1 to Equation (1) across the 74-year period on an annual mean basis. The major contribution in DJF (Figure 2a) occurred at latitudes of around 80° S, taking on approximately 80% of the variance in those years with a maximum (e.g., mid 1990s). It can also be seen that the contribution rapidly decreases towards the Poles, regardless of the year, whereas the reduction occurs in a gradual fashion towards lower latitudes and is year-dependent. On average, in the 50° S–70° S range, there is a reduction in wave 1 participation during the second half of the analyzed timespan. On the other hand, at latitudes between 30° S and 50° S, wave 1 seems to have a quasi-steady contribution. When compared to DJF, in MAM (Figure 2b), along 80° S, wave 1 has a more homogeneous

contribution across the years; in the 20° S–60° S range, its variance is lower, on average, across the entire period. The dominance of wave 1 along approximately 80° S is replicated in JJA (Figure 2c) and in SON (Figure 2d). When all the seasons are compared to each other, SON seems to be the one in which wave 1′s larger contributions extend towards lower latitudes. The disruption in wave 1′s participation between 1970 and 1980 in DJF is worth noting too, yet the causes are beyond the scope of this work. The contributions of waves 2–6 to (1), in terms of the represented variance, are shown in Figures S1–S5, respectively. Waves 2 and 3 (Figures S1 and S2, respectively) also had their major contributions (up to 40% and 25%, respectively) at high latitudes, but their zone of influence possessed a greater latitudinal extension, especially in the case of wave 3, whose activity centered around 60° S. Regarding wave 4, its contribution seemed to be more important (up to 30%) during MAM between 40° S and 60° S (Figure S3). The maximum represented variance of wave 5 showed a slight reduction when compared to wave 4 (25% against 30%), with a shift towards lower latitudes as well (Figure S4). Wave 6 made a contribution of up to 30% and was mostly restricted to latitudes lower than 50° S (Figure S5).

The construction of the BI for the entire SH followed the two criteria proposed in [14]. For the 144 longitude points, the index:

$$(BI)_i = z_i(\varphi = 35° \text{ S}) - z_i(\varphi = 50° \text{ S}) \quad (i = 1, \ldots, 144) \tag{2}$$

was evaluated using (1). Instead of retaining the first 18 harmonics, as is the case in [14], and considering the results described in the previous paragraph, we retained all the harmonics in (1), with a represented variance greater than 1% so that local effects through shorter waves could also be included. In addition to $(BI)_i < 0$ the condition:

$$\frac{(BI)_{i+4} + (BI)_i + (BI)_{i-4}}{3} < 0 \tag{3}$$

i.e., the index in (2) averaged over 30° of longitude, was also required so that single negative values were not included in the analysis. The simultaneous fulfillment of (2) and (3) excluded cases of 1-day episodes that had a small longitudinal extent. A blocking-finding algorithm was implemented on the daily 500 hPa geopotential height dataset to locate events for every 2.5° of longitude using a program written in FORTRAN. The software located 26261 individual cases at different longitudes that simultaneously satisfied conditions (2) and (3). In order to match our results to those of previous studies, these individual events were collected into pre-specified longitudinal bands of a width of 10°, in which boundaries were defined by the expression $lon(i) = 10° \times i \pm 5°$, with $i$ being an integer running from 0 to 36. The redefinition $lon(i) - 360°$ was implemented if $lon(i) > 180°$ in order to standardize the longitudes so that positive and negative values were associated with the Eastern Hemisphere (EH) and the Western Hemisphere (WH), respectively. For each individual located case with a generic longitude, *LON*, the condition $lon(i) \leq LON < lon(i+1)$ was checked so that the borderline events were not assigned to two different bands simultaneously.

An example that illustrates a particular situation for the 500 hPa circulation field is now presented before moving to the next section. Figure 3 shows the 500 hPa geopotential height in the SH, south of 20° S, during four consecutive dates from 29 June to 2 July 2007. On 29 June, 30 June, 1 July and 2 July, the foregoing blocking conditions were satisfied at all longitudes in the 110° W–92.50° W, 110° W–85° W, 110° W–82.50° W, and 110° W–85° W ranges, respectively. These blocking conditions persisted for these four days only. Figure 4 shows the time evolution of the BI for the 110° W–82.50° W sector for every 2.5° of longitude during the aforementioned period. This particular set of dates was chosen because on 1 July 2007, the BI reached the overall minimum value within the analyzed period (−266.83 m at 95° W). On this particular date, extreme values were also found at 100° W (−246.74 m), 97.50° W (−265.79 m), and 92.5° W (−249.24 m). The high-amplitude trough at 150° W in Figure 3a evolved into a cut-off low in Figure 3d. On the other hand, the

cut-off low that was centered approximately at 40° S and 110° W in Figure 3a experienced little change regarding both its position and its structure. The configuration of an omega blocking is visible in Figure 3d, between 150° W and 90° W. As for the contribution of the different zonal waves to the 500 hPa time series at the latitudes that define the BI, it is visually evident from the panels in Figure 3 that short waves played an important role during these dates. On 29 June, the most representative wave at 35° S was 1 (29.63%), followed by 7 (25.56%) and 6 (21.64%); at 50° S, waves 3 and 1 represented 40.59% and 22.82% of the variance, respectively. On 30 June, waves 1, 6, and 7 altogether accounted for almost 85% of the variance at 35° S, and the combination of waves 1, 2, 3, and 6 explained almost 80% of the variance at 50° S. On 1 July, waves 1, 6, and 7 accounted for 26.35%, 27.74%, and 19.56% of the variance at 35° S, respectively, whereas waves 1–3 altogether took on almost 71% of the variance and wave 6 represented another 11% at 50° S. Finally, on 2 July, waves 1, 5, 6, and 7 contributed 19.32%, 16.92%, 16.93%, and 20.27% to the variance at 35° S; wave 1 was by far the most representative (50.30%) at 50° S, followed by waves 2 (18.03%) and 3 (10.39%). Even though the establishment of persistent anomalies is generally driven by waves 1 and 3 [44], this example shows that shorter waves can also contribute on occasion.

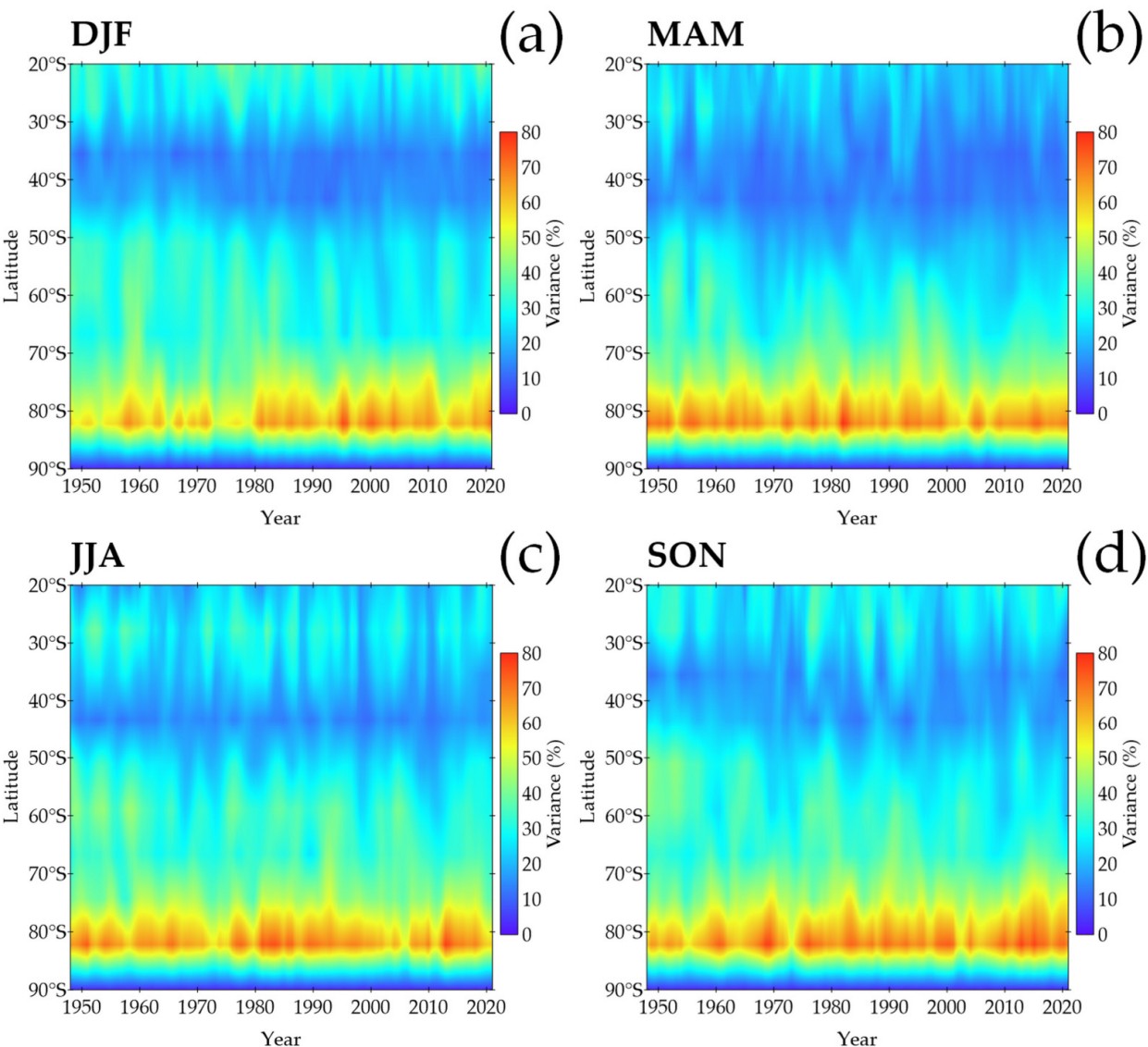

**Figure 2.** Contribution of wave 1 in Equation (1) to the total variance of the 500 hPa geopotential height across the 74-year period in terms of seasonal means: (**a**) DJF; (**b**) MAM; (**c**) JJA; (**d**) SON.

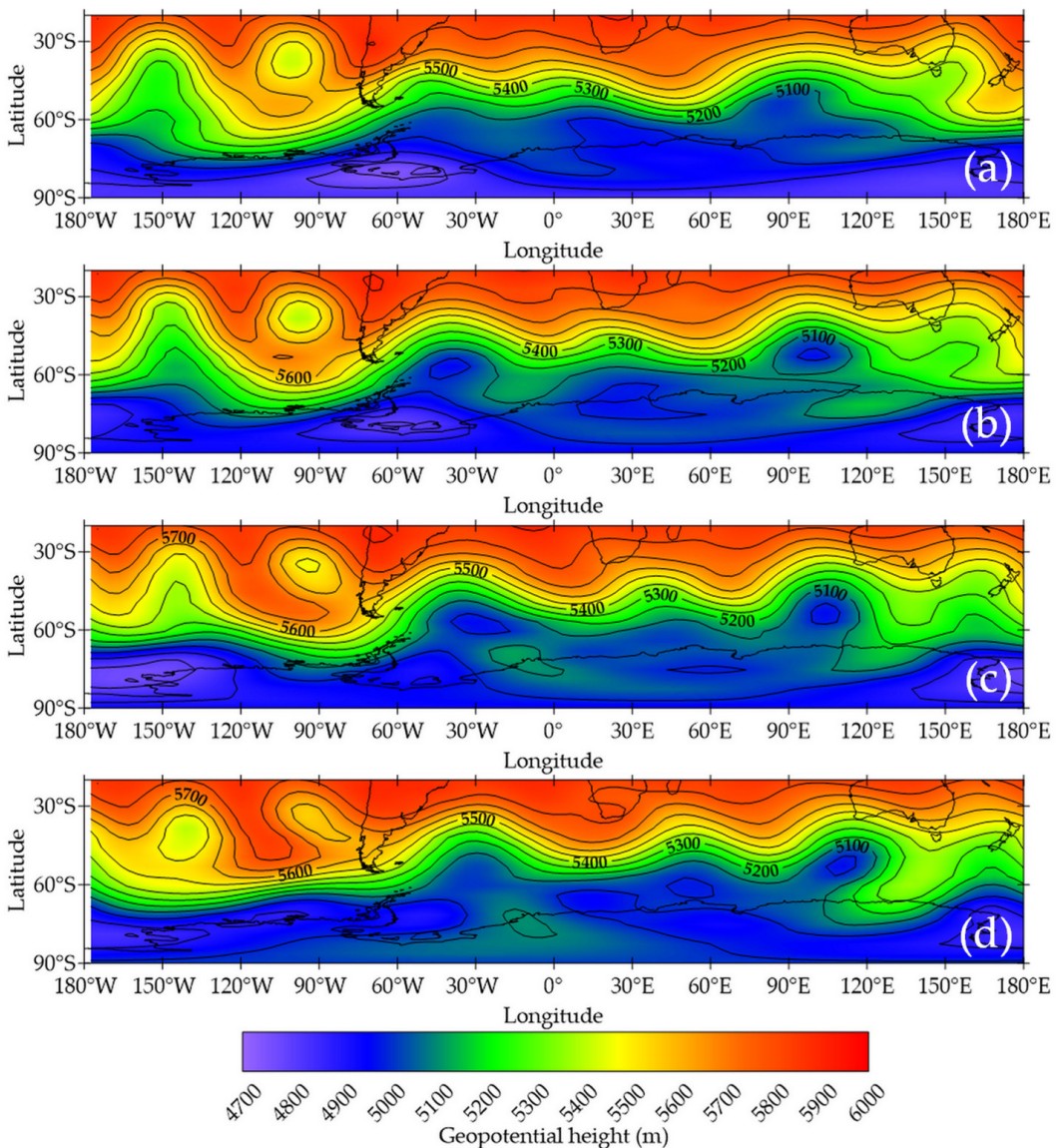

**Figure 3.** 500 hPa geopotential height in the SH, south of 20° S, as calculated from (1) over four consecutive days that fulfilled the blocking conditions at 95° W: (**a**) 29 June 2007; (**b**) 30 June 2007; (**c**) 1 July 2007; (**d**) 2 July 2007.

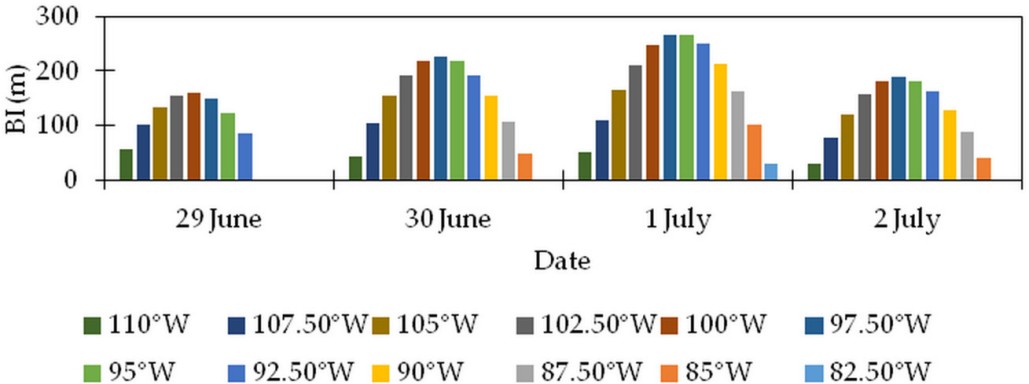

**Figure 4.** Time evolution of the absolute value of the BI between 29 June and 2 July 2007, for every 2.5° of longitude in the 110° W–82.50° W range. The BI on 1 July 2007 reached -266.83 m at 95° W, which was the overall maximum absolute value within the analyzed period. Values expressed in meters.

## 3. Results

*3.1. Distribution of Blockings Disregarding Their Duration*

### 3.1.1. Annual and Seasonal Distributions

Figure 5 shows the distribution of the blocking episodes regardless of their duration as a function of longitude at intervals of 10° for the annual case as well as for DJF, MAM, JJA, and SON. The vertical scale is logarithmic so that smaller values are highlighted. In order to assess on the symmetry of these distributions we made use of the skewness coefficient ($\gamma$) as defined in [46] (p. 28):

$$\gamma = \frac{\frac{1}{n-1} \sum_{i=1}^{n} \left( f_i - \overline{f} \right)^3}{s^3} \tag{4}$$

with $n$ being the total number of class intervals, $\overline{f}$ denoting the mean frequency calculated over all the class intervals, and $s$ representing the standard deviation of the frequencies across the $n$ intervals. The distributions depicted in Figure 5 were $n = 36$ and $\overline{f} = 2.78$. All these distributions had a $\gamma > 0$; therefore, blockings in the EH were more concentrated than in the WH. The annual distribution had a $\gamma = 0.95$ and shows the absolute maximum in the south Pacific Ocean at 160° W (9.34%), flanked by 170° W (9.21%) and 150° W (9.00%) (Figure 5a). As mentioned in the introduction, there is general agreement in the existing literature that the maximum occurs in the South Pacific (SP) region. However, the exact location and frequency of these occurrences are expected to be dependent upon the dataset from which the climatologies are extracted. For example, in [14], the authors utilized only eight years' worth of 500 hPa data from the Australian dataset; the maximum was also found in the SP region but west of the date line (DL, 180° W). Furthermore, the frequency of occurrence for the maximum in [14] was below 5%. There was a secondary maximum in SA at 40° W (0.99%), which evidences that, even though the annual frequency of blockings there was ten times smaller than those taking place to the east of the DL, SA was the second most important region in the SH in terms of annual occurrences. Although the frequency values between 10° W and 100° E are negligible when compared with the aforementioned values, they are high enough to consider the southern tip of Africa and its vicinities as another preferred region of blocking activity, as reported in previous research efforts [14,35,36].

When compared with the annual case, the maximum in DJF (13.98%) shifted to the west and sat along the DL (Figure 5b). In addition, there was an overall increase in the frequency of blockings between 140° E and 160° W and a decrease elsewhere. Actually, there was an overall increase in EH frequencies as the asymmetry increased to $\gamma = 1.56$ from $\gamma = 0.95$ in the annual distribution. This outcome is in qualitative agreement with [14]; the authors found greater frequencies around the SP region but for November and December only, whereas in January, there was a decrease with respect to the annual average. During MAM, almost all of the frequencies in the WH were higher than the annual average and the same occurred in the EH at most of the longitudes between the Greenwich Meridian (GM, 0°) and 80° E (Figure 5c). During this season the absolute maximum (9.71%) centered at 150° W; the secondary maximum in SA increased to 1.39% and took place at 40° W. The third-rank maximum, found in the Indian Ocean region, is in agreement with the existing literature [2] (p. 311). The distribution in this quarter was again right-tailed ($\gamma = 1.06$), yet this is visually less evident. In JJA, the maximum took place in the SP region at 150° W (9.01%) and was almost equal to its annual counterpart (9.00%) (Figure 5d). Since $\gamma = 0.75$, the distribution in this season was more symmetric than its annual counterpart. Almost all of the longitudes in the WH east of the maximum experienced an increase in their frequencies with respect to the annual average. By contrast, there was a notable reduction in the boreal frequencies with respect to the annual average in the 170° E–160° W sector. There was a regain in the frequencies west of 170° E, with values above the annual average. These results are partly in concordance with [14], for they found, particularly in June, a remarkable decrease in the frequencies around the DL, which was surrounded by two

maxima that exceeded the annual averages. As in MAM and JJA, in SON, the absolute maximum (9.27%) also took place east of the DL (Figure 5e). This maximum was flanked by 180° E and 160° W, with the latter longitude having a frequency below the annual average. This distribution had $\gamma = 0.93$. Overall, the springtime frequencies in the WH were greater than that of the annual averages. In particular, the second-rank maximum during the season (1.69%) was located east of SA at 40° W. In contrast to the WH, in the EH, there was an overall reduction in the frequency with respect to the annual average. The results are qualitatively in agreement with the ones presented in [14], particularly those related to the southeastern SA sector.

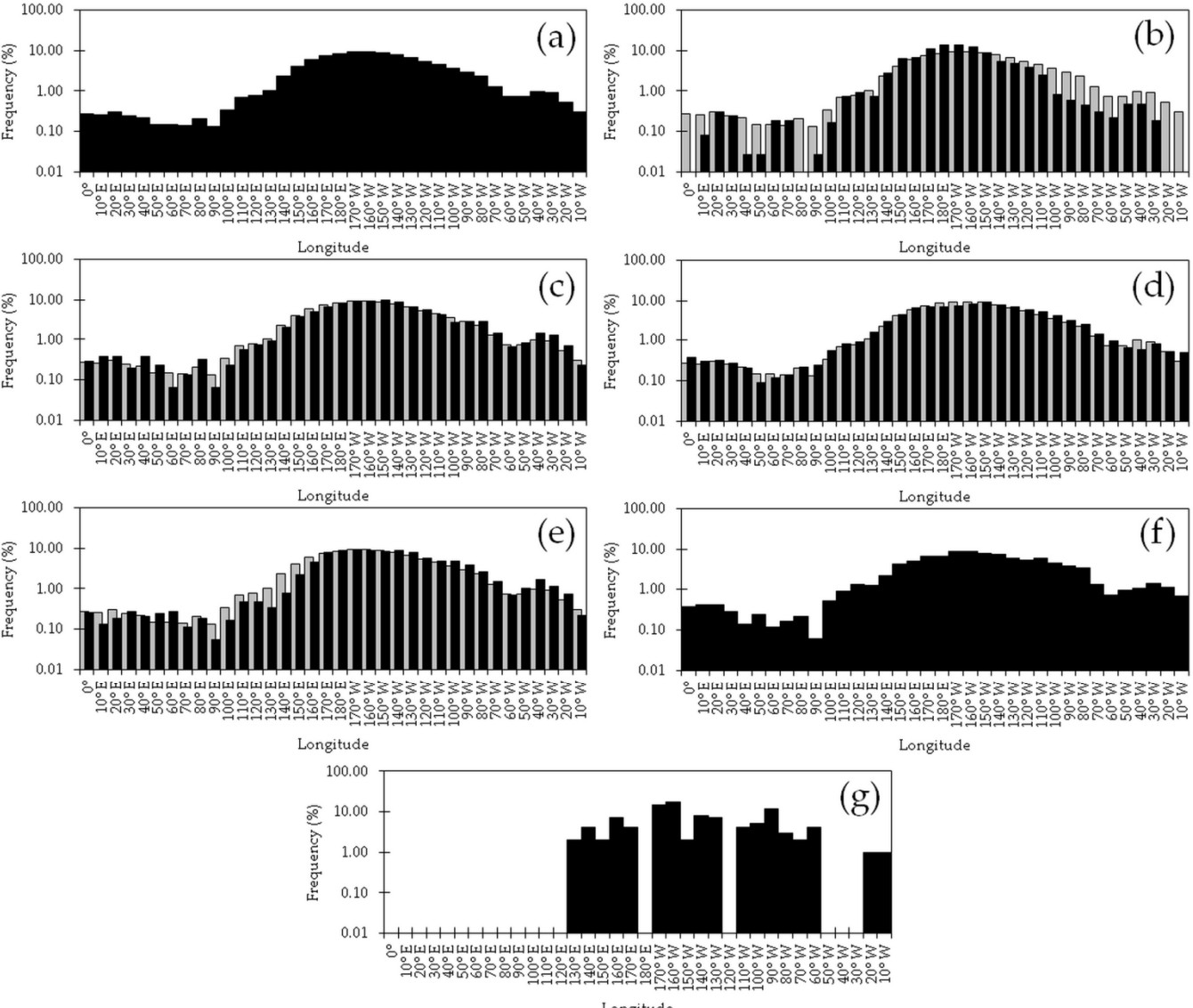

**Figure 5.** Longitudinal distribution of blocking episodes at 500 hPa within the 74-year period: (**a**) annual; (**b**) DJF; (**c**) MAM; (**d**) JJA; (**e**) SON. The vertical scale is logarithmic. The total number of days was 27,010, 6660, 6808, 6808, and 6734 for the annual, DJF, MAM, JJA, and SON cases, respectively. The grey bars in panels (**b**–**e**) represent the annual statistics in (**a**) and are presented for comparison purposes. Shown in (**f**,**g**) are the annual frequencies for values of the BI below −100 m and −200 m, respectively. All the frequencies are relative to the corresponding panel's totals. Individual dates were counted regardless of the duration of the episodes. The skewness coefficients of these distributions are: (**a**) 0.95; (**b**) 1.56; (**c**) 1.06; (**d**) 0.75; (**e**) 0.93; (**f**) 0.77; (**g**) 1.89.

Figure 5a–e involved $BI < 0$ without making distinctions among the absolute values of the BI, which can be considered as a measure of the strength of an event. Shown in Figure 5f,g are the annual distributions of the BI across the longitudes for $BI < -100$ m and $BI < -200$ m, respectively, i.e., stronger longitudinal gradients. The distribution of $BI < -100$ m (Figure 5f) is similar to the one for $BI < 0$ (Figure 5a) in the sense that it had the maximum values in the SP region. However, the maximum (8.67%) occurred at 170° E, and there was a secondary maximum (5.76%) that also took place in the SP region at 110° W. Given that $\gamma = 0.77$ for Figure 5f, the distribution of blockings with $BI < -100$ m was more homogeneous across the longitudes than in the $BI < 0$ case. It is worth noting that the frequencies around the southern tip of Africa and to the east of SA are greater than their counterparts in Figure 5a. In other words, when compared with all the possible strengths compatible with a blocking, those events with intermediate ($BI < -100$ m) and extreme ($BI < -200$ m) values had the potential of greater occurrences in these two regions. Nonetheless, when only extreme events were considered (Figure 5g), they were restricted to four different regions, namely 130° E–170° E, 170° W–130° E, 110° W–60° W, and 20° W–10° W; they accounted for 19%, 49%, 30%, and 2% of the extreme cases, respectively. Hence, only mild and intermediate events took place in the 0°–120° E sector. On the other hand, the SP region east of the DL had, on average, not only the greatest count of blocking events but also the strongest ones. The distribution of $BI < -200$ m was the most asymmetric one ($\gamma = 1.89$) among those presented in Figure 5.

Before proceeding to the analysis of the monthly distributions, it is helpful to know the way in which the zonal waves in (1), for the latitudes that defined the BI, i.e., 35° S and 50° S, combined on each of the dates when the blocking events occurred. Table 1 shows the annual count of the combination of the waves that had the single maximum represented variance for each of these two latitudes within the entire 74-year period. In order to avoid over counting, specific dates on which the condition $BI < 0$ and the one in (3) were simultaneously satisfied in more than one of the 144 original longitudes were counted once. The number of dates in the annual counting was 3761. If we use the labeling "wave at 35° S–wave at 50° S" to denote each combination, the more frequent pair resulted in the 1–1 one, which represented 17.70% of the cases, followed by the 1–3 (9.57%) and the 1–4 (6.64%) pairs. It is interesting to note that for an annual average, blockings on 1501 dates (39.90% of the total) occurred with wave 1 representing the maximum variance at 35° S. On the other hand, wave 1, having its maximum variance at 50° S, totaled 1674 dates (44.51%).

**Table 1.** Annual counting of the combination of the zonal waves that had the single maximum represented variance at 35° S and 50° S on the dates when blocking conditions were fulfilled at least in 1 of the 144 analyzed longitudes. The number of dates totaled 3761.

| 35° S | 50° S | | | | | | Total |
|---|---|---|---|---|---|---|---|
| | 1 | 2 | 3 | 4 | 5 | 6 | |
| 1 | 666 | 159 | 360 | 250 | 57 | 9 | 1501 |
| 2 | 209 | 74 | 109 | 86 | 19 | 2 | 499 |
| 3 | 141 | 48 | 76 | 40 | 8 | 2 | 315 |
| 4 | 217 | 61 | 105 | 110 | 7 | – | 500 |
| 5 | 232 | 69 | 108 | 58 | 30 | 1 | 498 |
| 6 | 145 | 20 | 78 | 40 | 14 | – | 297 |
| 7 | 52 | 11 | 37 | 13 | 4 | – | 117 |
| 8 | 11 | 4 | 8 | 5 | 3 | – | 31 |
| 9 | 1 | 1 | 1 | – | – | – | 3 |
| **Total** | 1674 | 447 | 882 | 602 | 142 | 14 | 3761 |

As for DJF and to JJA (results not shown), the total count of dates was 571 and 1444, respectively, which accounted for 15.18% and 38.39% of the total number of dates for the annual average, respectively. These figures reflect the fact that JJA is the season with the greatest number of blockings. During both DJF and JJA, the most frequent combination of waves was the 1–1 pair, which represented 15.58% and 20.43% of the cases, respectively.

Overall, in DJF, wave 1 had the maximum represented variance at 35° S and 50° S in 157 (27.50%) and 316 (55.34%) of the cases, respectively. In JJA, these figures increased to 737 (51.04%) and 582 (40.30%), respectively. A notable difference between DJF and JJA is that the second most representative wave combination was the 1–3 in JJA (14.00%), whereas in DJF, this was wave 1 at 50° S combined with waves 2, 4, 5, and 6 at 35° S in an quasi-evenly fashion, altogether representing 33.98% of the cases. The results are in good agreement with the analysis carried out on gridded 500 hPa data by the authors of [44], which revealed that zonal wave 1 plays a fundamental role in the dynamics of blockings.

### 3.1.2. Monthly Distributions

The annual and seasonal distributions of the BI presented in Figure 5 made no distinctions between the different months that comprise the seasons. Such separation is made in Figure 6, which shows the monthly distribution of the frequency of detected blocking episodes within the 74-year period regardless of the longitude (Figure 6a) and for every 20° of longitude (Figure 6b–s). Also shown are the skewness coefficients. Although less helpful than in the analysis of Figure 5, this serves as a tool for identifying whether blockings tended to take place in any particular season. For the distributions in Figure 6, the values of $n$ and $\overline{f}$ in Equation (4) were 12 and 8.33, respectively.

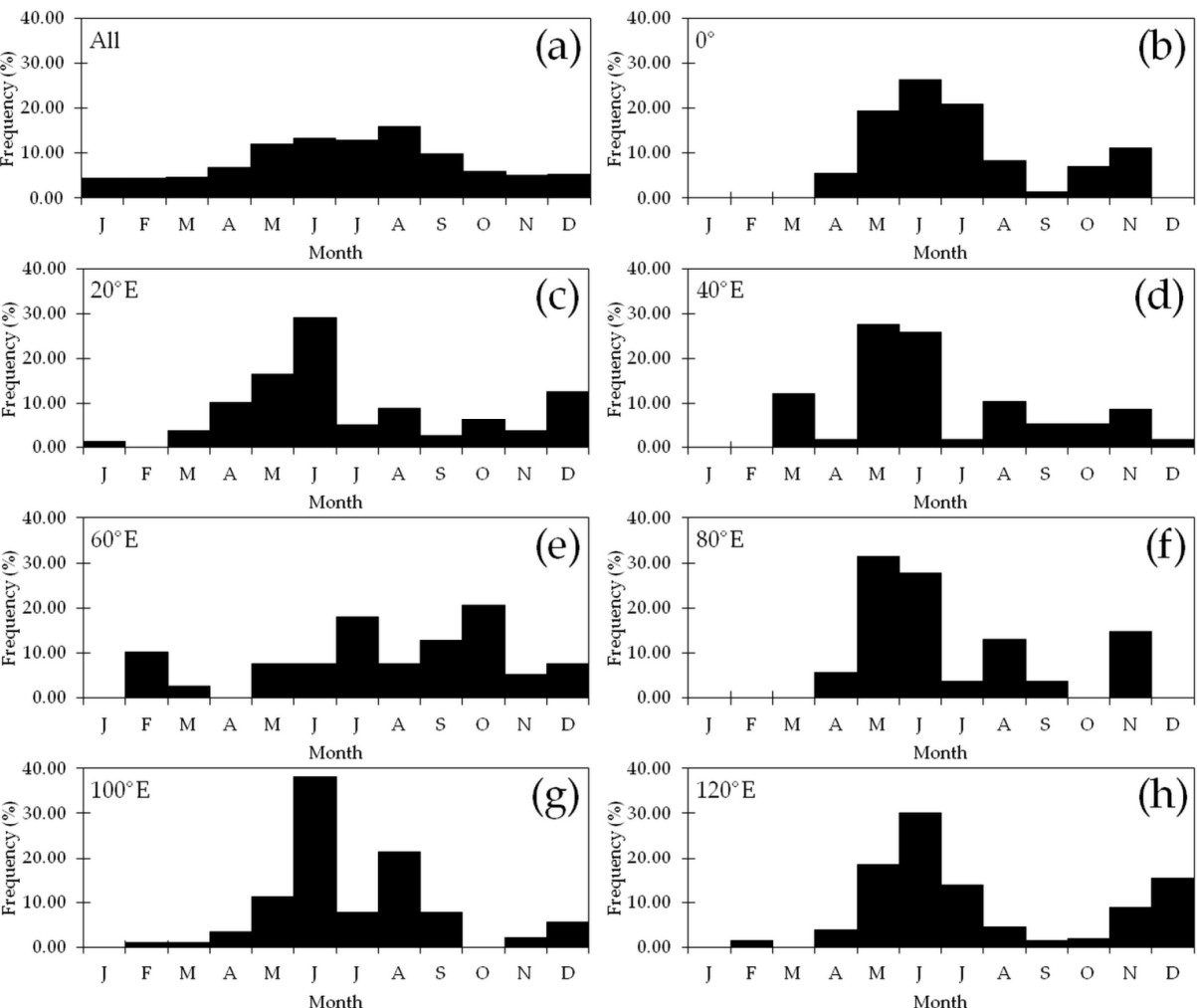

**Figure 6.** *Cont.*

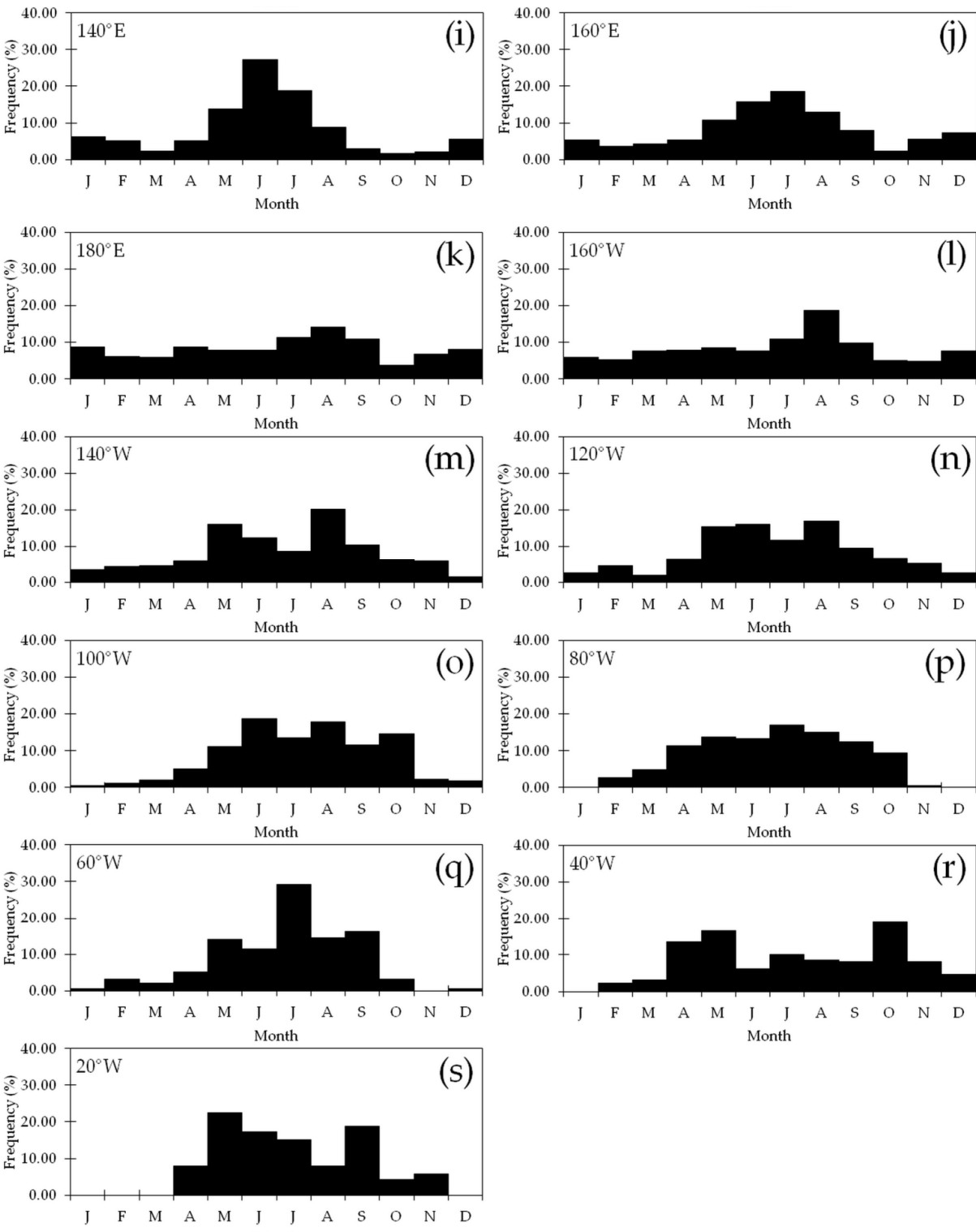

**Figure 6.** Monthly evolution of the frequency of blocking episodes within the 74-year study period at different longitudes: (**a**) All longitudes; (**b**) 0°; (**c**) 20° E; (**d**) 40° E; (**e**) 60° E; (**f**) 80° E; (**g**) 100° E; (**h**) 120° E; (**i**) 140° E; (**j**) 160° E; (**k**) 180° E; (**l**) 160° W; (**m**) 140° W; (**n**) 120° W; (**o**) 100° W; (**p**) 80° W; (**q**) 60° W; (**r**) 40° W; (**s**) 20° W. All negative values of the blocking index were included in the calculations regardless of their absolute value. The skewness coefficients of these distributions are: (**a**) 0.54; (**b**) 0.35; (**c**); 1.39 (**d**) 0.96; (**e**) 0.70; (**f**) 0.51; (**g**) 1.53; (**h**) 0.92; (**i**) 1.33; (**j**) 0.77; (**k**) 0.47; (**l**) 1.73; (**m**) 0.88; (**n**) 0.42; (**o**) 0.21; (**p**) −0.55; (**q**) 0.94; (**r**) 0.53; (**s**) 0.16.

Figure 6a shows that blockings in the SH were present throughout the year. Occurrences above 10% of the total number of detected individual events took place between May and August, i.e., late autumn and mid-winter, with the absolute maximum occurring in the latter month (15.82%). On the other hand, the minimum frequencies happened in the summertime, and it was in January that the absolute minimum took place (4.36%). Even though the distribution was centered in the winter months, it was not symmetric. By virtue of $\gamma = 0.54$ for the distribution, the autumn-to-winter transition was steeper than the winter-to-spring shift. Although a detailed analysis of the differences across the longitudes (Figure 6b–s) is not the scope of the present work, it is worth noting that the most homogeneous distribution, in the sense that the frequencies for all the months were closer to equiprobability, occurred at 180° E (yet $\gamma = 0.47$, so the greater frequencies located in the first half of the year) (Figure 6k). In other words, it was at 180° E where blockings took place in a more even fashion all year long, in contrast to other longitudes where the phenomena were much more concentrated on a seasonal basis (e.g., 20° W with no cases at all in summer, Figure 6s). At 180° E, the maximum frequency of blockings was in August, which accounted for 14.06% of the total number of cases located at this particular longitude. By contrast, October, which was the month with the minimum number of episodes, accounted for just 3.81% of the cases. In terms of the homogeneity of the distribution across the year, 180° E seemed to be followed by 160° E (Figure 6j), with a maximum of 18.54% (August) versus a minimum of 2.32% (October). The minima in the distributions of these two latitudes were concurrent, whereas the maxima were out of phase by a month. In contrast, it was at 100° E (southeastern Indian Ocean) where the greatest contrasts took place, with July having 38.20% of the total when compared with months with no cases at all (January and October) (Figure 5g).

### 3.2. Duration
#### 3.2.1. By Longitude

No distinctions were made so far between the duration of the located episodes. Shown in Figure S6 is an animation including the annual and the seasonal distributions of the detected blocking episodes for every 10° of longitude separated by days of persistence (1 day, 2–4 days, and ≥5 days, paralleling [14]) and by season (annual, DJF, MAM, JJA, and SON). All the frequencies were calculated relative to the total number of events accounted for in the corresponding panel of duration (not to the overall count of the cases in the season, which is dealt with below) so that all the distributions are comparable to each other. The vertical scale is logarithmic, as in Figure 5. For the persistence of two or more days, the date was assigned to the season wherein the blocking episode was first detected. The annual distribution of 1-day persistence events was concentrated around 170° W, where the maximum (8.67%) took place; a second-rank maximum occurred southeast of SA at 40° W (1.28%) (Figure S6a). The annual distribution of persistences of 2–4 days was also concentrated around 170° W (maximum of 9.98%) and the secondary maximum also occurred southeast of SA at 30° W (0.73%) (Figure S6b). Episodes with a persistence ≥5 days were almost exclusively concentrated in the SP region (99.90% of the cases), and their distribution shifted to the east when compared with the former two counterparts, with a maximum of 17.17% at 150° W; the other longitude value that had events with such persistence took place at 20° W (Figure S6c). In DJF, 1-day persistence episodes more frequently sat at 180° E (13.20%) (Figure S6d). It is worth noting that the number of cases in the surroundings of 180° E during the season was remarkably greater than the annual average. There was a steeper transition of the frequencies from above to below the annual average in the WH, and the latitude at which this took place was located somewhere between 150° W and 140° W (Figure S6e). Regarding the events with 2–4 days of persistence, they primarily occurred at 170° W (15.17%) and its vicinities with frequencies greater than the annual average too (Figure S6f). The occurrence of events with 5 or more days of persistence during the season was restricted to the SP region at 150° E, and in

the 170° E–150° W and 130° W–110° W ranges, with a maximum at 130° W (20.34%), considerably greater than the annual average (Figure S6f).

Unlike DJF, during MAM, the single-day blockings' frequencies were closer to the annual average at most of the longitudes. One of the most notable exceptions, with an above-average frequency, was at 40° E, where the cases there almost doubled (0.68% versus 0.38%). On the other hand, even though the maximum occurred at 160° W (8.96%) and was above the annual average, in the SP region, the frequencies were below the annual average for most of the longitudes (Figure S6g). In the same region, events with a duration of 2–4 days were more frequent than in the annual average for the 150° W–110° W belt (actually, the maximum of the distribution was 10.87% at 140° W). Other above-average frequencies beyond the SP region occurred between 90° W and 30° W and in the 0°–20° E range (Figure S6h). In contrast to DJF, events lasting 5 days or more in MAM did occur beyond the SP area, yet the 150° W–130° W range accounted for 57.33% of the cases (versus 43.31% in the annual average). The absolute maximum took place at 150° W (19.83%). Another area worth considering was the 90° W–80° W belt (south-western SA), where these particular cases more than tripled the annual average (they altogether accounted for 8.19% of the cases versus 2.60%) (Figure S6i).

In JJA, the number of 1-day events between 170° E and 150° W decreased with respect to the annual figures, yet the maximum was reached at 150° W (7.56%). By contrast, the frequency of 1-day long episodes for the rest of the longitudes within the SP region was greater than the annual average, as it was also in south-western SA (Figure S6j). As for the 2–4-day events, in the SP region, there was a decrease with respect to the annual average within an area circumscribed by 170° E and 130° W. The frequencies of these events in the SP region beyond this sector were greater than the annual average, as too were both the south Atlantic Ocean and in the eastern Indian Ocean (Figure S6k). Even though the events with a persistence of ≥5 days, that were present in the SP region, represented the majority of these episodes during the season (99.83% of the cases, with the maximum of 17.91% taking place at 150° W), this is the only season in which there was also a small number of them (0.17% of the cases) in the South Atlantic at 20° W (Figure S6l).

The distribution of 1-day events in SON was similar to the MAM counterpart, with a distinction that there was an overall decrease in the number of episodes west of 180° E in favor of a general increase to the east of this longitude (Figure S6m). The distribution of 2–4-day events in SON also resembled the MAM counterpart but with a reduced number of episodes in south-eastern Africa (Figure S6n). During SON, the ≥5-day events had two regions of occurrence, namely the 160° E–150° W and 130° W–90° W sectors that represented 64.71% and 35.29% of the cases, respectively, showing a net increase with respect to the annual average (49.40% and 34.93%, respectively) in both belts (Figure S6o).

It is considered that the interplay between the large-scale flow and synoptic eddies has a relevant role in the occurrence and the maintenance of blockings ([45] and references therein). A three-dimensional instability theory on the onset of blocking and cyclogenesis showed that one of the fastest-growing modes that solved the equations of a two-layer spherical quasigeostrophic model consisted of blocking-like structures located at favored regions [47]. A common feature of all the distributions in Figure S6 was that the primary region for blockings was the SP region. As for the 1-day and the 2–4-day distributions, these were similar regarding the location of the preferred formation regions. By comparing these two distributions with the ≥5-day counterpart, it can be assumed that the formation of most of the events was likely driven by conditions similar to the ones that took place at particular regions as described in [47], whereas their duration depended upon additional requirements, such as an increase or decrease in the static stability. The inclusion of events with lesser durations is hence warranted.

### 3.2.2. By Region

Table 2 shows the distribution of blocking episodes stratified by duration, season, and area of occurrence for the analyzed 74-year period. In order to parallel our outcome

to previous studies, three different regions were defined: the Pacific region (PAC), the Atlantic region (ATL), and the Indian Ocean region (IND), spanning the 110° E–80° W, the 70° W–0° W, and the 10° E–100° E ranges, respectively. Please note that these regions do not cover the entire SH as there are gaps of 10° in longitude that separate them. Unexpectedly, and in agreement with [14], the highest number of blocking episodes on an annual basis corresponded to the 1-day events. The majority of them occurred in JJA (5375, or 39.42%), whereas the minority happened in DJF (2017, 14.79%). In agreement with all the figures presented so far, the annual and the seasonal numbers show that most of the 1-day episodes took place somewhere in the PAC, followed by the ATL and the IND. As for the 2-day episodes, their total count was 6905, with the majority and minority of them taking place in JJA (2841, 41.14%) and in DJF (1051, 15.22%), respectively. Unlike with 1–2-day cases, for ≥3-day-long episodes, the occurrences were restricted to seasons, regions, or both. For instance, no 3-day events were recorded in the ATL during DJF or in the IND during SON. As a matter of fact, the percentages of blocking episodes in the ATL and in the IND decreased in favor of an increase of those that took place in the PAC as the duration increased. The longest detected event persisted for 12 days, in contrast to [14], where the authors found events that persisted for up to 13 days (yet no 11- or 12-day-long episodes were found by them). By permitting the BI to be a small positive, it was found in [30] that, by using the same dataset as is used in the present study, the maximum persistence in the SH was 26 days, evidencing that the results are highly dependent upon the conditions applied to the data. The longest detected events found in the ATL, and in the IND persisted for 5 and 4 days, respectively, and all of them occurred in JJA. In these particular cases, there was a single detected 5-day event in the ATL and there were five 4-day events located in the IND. Blocking episodes lasting 6 to 12 days were found exclusively in the PAC, in close concordance with [14], where the authors found the same for the range of 7 to 13 days. There were only 5 and 3 11- and 12-day-long events detected there, respectively; all of them occurred in JJA.

**Table 2.** Number of episodes and percentages of detected blockings with durations spanning from 1 to 12 days within three different regions: PAC, ATL, and IND (see text for further details). The figures are discriminated by season and correspond to the entire analyzed 74-year period. Seasons with no registered events were omitted.

| Duration | Season | Number of Episodes | | | | % of All Episodes | | |
|---|---|---|---|---|---|---|---|---|
| | | Total | PAC | ATL | IND | PAC | ATL | IND |
| **1** | Annual | 13,633 | 12,385 | 870 | 378 | 90.85 | 6.38 | 2.77 |
| | DJF | 2017 | 1943 | 43 | 31 | 96.33 | 2.13 | 1.54 |
| | MAM | 3248 | 2880 | 258 | 110 | 88.67 | 7.94 | 3.39 |
| | JJA | 5375 | 4894 | 320 | 161 | 91.05 | 5.95 | 3.00 |
| | SON | 2993 | 2668 | 249 | 76 | 89.14 | 8.32 | 2.54 |
| **2** | Annual | 6905 | 6475 | 338 | 92 | 93.77 | 4.90 | 1.33 |
| | DJF | 1051 | 1029 | 14 | 8 | 97.91 | 1.33 | 0.76 |
| | MAM | 1618 | 1506 | 88 | 24 | 93.08 | 5.44 | 1.48 |
| | JJA | 2841 | 2644 | 153 | 44 | 93.07 | 5.39 | 1.55 |
| | SON | 1395 | 1296 | 83 | 16 | 92.90 | 5.95 | 1.15 |
| **3** | Annual | 2983 | 2892 | 64 | 27 | 96.95 | 2.15 | 0.91 |
| | DJF | 422 | 418 | — | 4 | 99.05 | — | 0.95 |
| | MAM | 716 | 692 | 18 | 6 | 96.65 | 2.51 | 0.84 |
| | JJA | 1362 | 1306 | 39 | 17 | 95.89 | 2.86 | 1.25 |
| | SON | 483 | 476 | 7 | — | 98.55 | 1.45 | — |
| **4** | Annual | 1259 | 1236 | 18 | 5 | 98.17 | 1.43 | 0.40 |
| | DJF | 141 | 141 | — | — | 100.00 | — | — |
| | MAM | 303 | 298 | 5 | — | 98.35 | 1.65 | — |
| | JJA | 640 | 622 | 13 | 5 | 97.19 | 2.03 | 0.78 |
| | SON | 175 | 175 | — | — | 100.00 | — | — |

**Table 2.** *Cont.*

| Duration | Season | Number of Episodes | | | | % of All Episodes | | |
|---|---|---|---|---|---|---|---|---|
| | | Total | PAC | ATL | IND | PAC | ATL | IND |
| **5** | Annual | 530 | 529 | 1 | – | 99.81 | 0.19 | – |
| | DJF | 40 | 40 | – | – | 100.00 | – | – |
| | MAM | 141 | 141 | – | – | 100.00 | – | – |
| | JJA | 288 | 287 | 1 | – | 99.65 | 0.35 | – |
| | SON | 61 | 61 | – | – | 100.00 | – | – |
| **6** | Annual | 237 | 237 | – | – | 100.00 | – | – |
| | DJF | 9 | 9 | – | – | 100.00 | – | – |
| | MAM | 63 | 63 | – | – | 100.00 | – | – |
| | JJA | 138 | 138 | – | – | 100.00 | – | – |
| | SON | 27 | 27 | – | – | 100.00 | – | – |
| **7** | Annual | 119 | 119 | – | – | 100.00 | – | – |
| | DJF | 4 | 4 | – | – | 100.00 | – | – |
| | MAM | 24 | 24 | – | – | 100.00 | – | – |
| | JJA | 73 | 73 | – | – | 100.00 | – | – |
| | SON | 18 | 18 | – | – | 100.00 | – | – |
| **8** | Annual | 62 | 62 | – | – | 100.00 | – | – |
| | DJF | 4 | 4 | – | – | 100.00 | – | – |
| | MAM | 2 | 2 | – | – | 100.00 | – | – |
| | JJA | 49 | 49 | – | – | 100.00 | – | – |
| | SON | 7 | 7 | – | – | 100.00 | – | – |
| **9** | Annual | 30 | 30 | – | – | 100.00 | – | – |
| | DJF | 2 | 2 | – | – | 100.00 | – | – |
| | JJA | 24 | 24 | – | – | 100.00 | – | – |
| | SON | 4 | 4 | – | – | 100.00 | – | – |
| **10** | Annual | 10 | 10 | – | – | 100.00 | – | – |
| | JJA | 8 | 8 | – | – | 100.00 | – | – |
| | SON | 2 | 2 | – | – | 100.00 | – | – |
| **11** | JJA | 5 | 5 | – | – | 100.00 | – | – |
| **12** | JJA | 3 | 3 | – | – | 100.00 | – | – |

*3.3. Trends*

Annual and seasonal time series of the BI and of the number of individual episodes were constructed for every 10° of longitude in order to estimate their linear trends. The minimum number of values that were required to calculate the BI averages was three, otherwise, the annual or seasonal average was flagged as missing. Missing values were not considered in the trend calculations. A minimum number of five elements in each time series was imposed in order to estimate the trends. Table 3 shows the results of the trend analysis at longitudes where significant values of the slope were detected either for the BI time series or for the time series of the number of episodes.

Regarding the BI time series, downward trends were detected in the annual series at 140° E, 180° E, and 70° W; the rate of decrease of the mean BI in these cases was −55, −23, and −53 m century$^{-1}$, respectively, meaning that stronger episodes were, on average, present at the end of the series at these longitudes. Mixed results were found for DJF, as a downward trend of −110 m century$^{-1}$ was detected at 140° E, and upward trends (i.e., weaker episodes at the end points of the series) were located at 170° E and 150° W (46 and 31 m century$^{-1}$, respectively). Negative trends were detected at a single longitude for the rest of the seasons: −136 m century$^{-1}$ at 130° E (MAM), −49 m century$^{-1}$ at 140° E (JJA), and −57 m century$^{-1}$ at 180° E (SON). As for the number of episodes, positive trends in the annual averages (i.e., more blocking episodes detected at the end of the series) were located at 150° E, 170° E, 180° E, and 170° W, with an increase in their frequency at a rate of 19, 20, 33, and 33 episodes century$^{-1}$, respectively. On the other hand, a single negative trend was located at 130° W (−21 episodes century$^{-1}$). Seasonally, the DJF and JJA time series had a single detected trend in each: in the former case at 110° W (−19 episodes century$^{-1}$) and in the latter case at 10° W (10 episodes century$^{-1}$). MAM had trends of

$-14$ and $-11$ episodes century$^{-1}$ at 120° W and 110° W, respectively. Finally, SON had trends at three different longitudes: 150° W, 50° W, and 30° W with slopes of $-18$, $-7$, and 9 episodes century$^{-1}$, respectively. The results concerning the annual number of episodes are in contradiction with those presented in [27] since they found a downward trend for the whole of the SH, yet they used a somewhat different blocking criteria. The authors did not analyze the seasonal values.

**Table 3.** Linear trends in the annual and seasonal time series of the BI and of the number of individual episodes for every 10° of longitude. All trends shown are significant to a 95% confidence level. The number of elements in each time series is shown in parenthesis. Longitudes at which no significant trends were detected are omitted.

| Longitude | Trend | | | | | | | | | |
| --- | --- | --- | --- | --- | --- | --- | --- | --- | --- | --- |
| | BI (m Century$^{-1}$) | | | | | Number of Episodes (Number Century$^{-1}$) | | | | |
| | Season | | | | | Season | | | | |
| | Annual | DJF | MAM | JJA | SON | Annual | DJF | MAM | JJA | SON |
| 130° E | – | – | $-136$ (10) | – | – | – | – | – | – | – |
| 140° E | $-55$ (47) | $-110$ (13) | – | $-49$ (31) | – | – | – | – | – | – |
| 150° E | – | – | – | – | – | 19 (56) | – | – | – | – |
| 170° E | – | 46 (32) | – | – | – | 20 (70) | – | – | – | – |
| 180° E | $-23$ (72) | – | – | – | $-57$ (39) | 33 (72) | – | – | – | – |
| 170° W | – | – | – | – | – | 33 (70) | – | – | – | – |
| 150° W | – | 31 (29) | – | – | – | – | – | – | – | $-18$ (33) |
| 130° W | – | – | – | – | – | $-21$ (62) | – | – | – | – |
| 120° W | – | – | – | – | – | – | – | $-14$ (28) | – | – |
| 110° W | – | – | – | – | – | – | $-19$ (7) | $-11$ (27) | – | – |
| 70° W | $-53$ (34) | – | – | – | – | – | – | – | – | – |
| 50° W | – | – | – | – | – | – | – | – | – | $-7$ (8) |
| 30° W | – | – | – | – | – | – | – | – | – | 9 (8) |
| 10° W | – | – | – | – | – | – | – | – | 10 (7) | – |

*3.4. A Radiosonde-Derived Blocking Index*

In this section, we calculated the blocking index in southern SA (SSA) using alternative data. One of the motivations for focusing the attention of the study to blockings in SSA is that the strong south-easterly winds in the Rio de la Plata, the so-called *sudestadas*, partly owe their existence to the presence of low-over-high-associated cut-off lows in the north of the Buenos Aires province [48] (p. 188) [49]. The upper-air network in the SH is sparse for many practical purposes, but we were able to take advantage of the location of two operational radiosonde stations along approximately 58.50° W to construct an alternative blocking index for this longitude. These two upper-air stations are Ezeiza (34.81° S, 58.53° W) in Argentina and Mount Pleasant (51.81° S, 58.45° W) in the Falkland Islands (Malvinas). According to the World Meteorological Organization, the five-number identifiers for these two stations are 87,576 and 88,889, respectively. The 500 hPa radiosonde dataset for these two stations was obtained from the University of Wyoming worldwide radiosonde database [50]. Radiosondes in this database are available from 1973 for the case of Ezeiza and from 1988 for the case of Mount Pleasant. The construction of the RBI required simultaneous values from the 500 hPa geopotential height at both locations. During the 1988–2021 period, there were 9057 pairs of radiosondes launched either at 00Z or at 12Z. The RBI was constructed by subtracting the 500 hPa geopotential height at Ezeiza from the corresponding height at Mount Pleasant and by selecting the negative values. The subtraction resulted in a negative value in only 84 cases. These RBIs were correlated with the BIs at the original longitudes. If there were two values of the RBI on the same date, i.e., one at 00Z and another one at 12Z, only the first one was included in the calculations of the correlation. More specifically, given that for eight particular dates of the 84 cases, there were radiosonde launchings at both 00Z and 12Z, the net number of individual RBIs available

for the correlations was 74. Figure 7 shows the composite of the 500 hPa fields in SSA over these 74 dates. This composite is in agreement with a blocked flow in the area, as it shows a cyclonic (anticyclonic) circulation in the northern (southern) portion of the domain, in resemblance with a "low-over-high" blocking. In between this, the anticyclonic circulation that protruded into the southern tip of SA at approximately 45° S was complemented to the northeast with a closed cyclonic circulation in a cut-off stage. Even though condition (3) was not checked for the RBI, this index proves to be suitable for the analysis of blockings in the region since the entire system in the composite spans at least 40° of longitude.

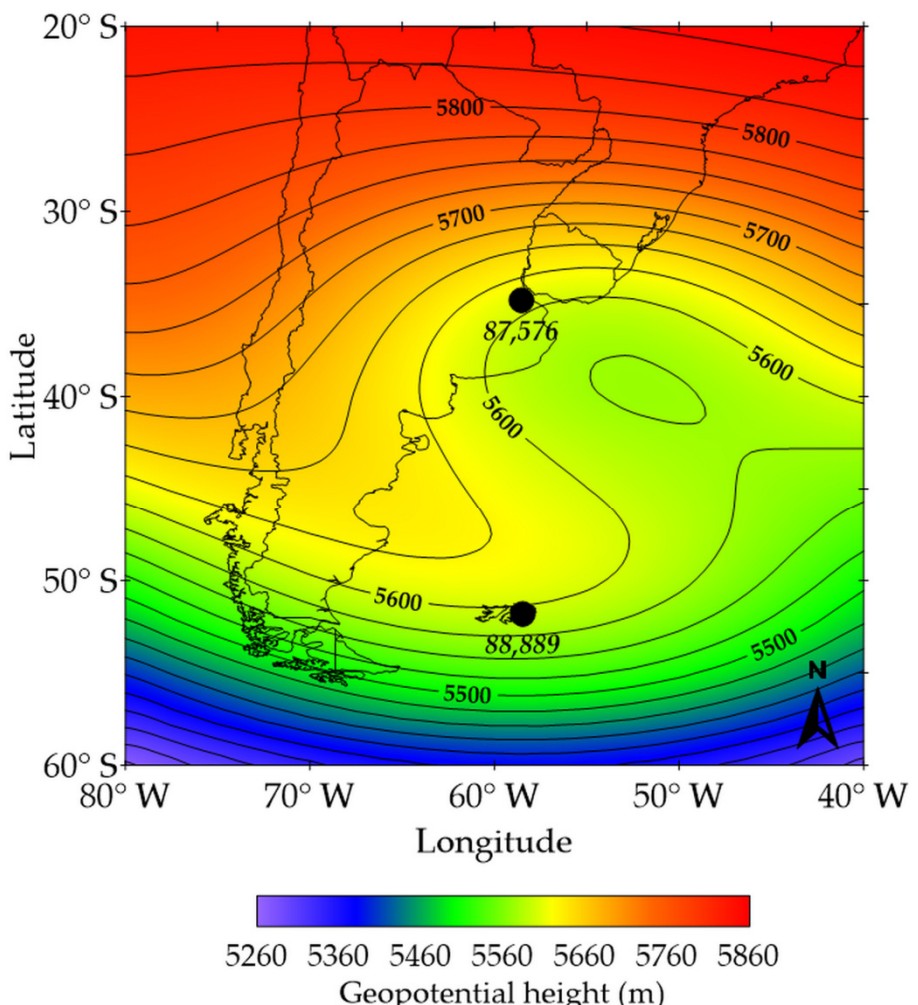

**Figure 7.** Composite of the 500 hPa fields in SSA, including the 74 dates on which there was at least an RBI. The black dots show the location of the upper-air stations used in the calculation of the RBI. See text for further details.

Table 4 shows the Pearson ($r_P$) and the Spearman ($r_S$) correlation coefficients between the RBI and the BI at selected longitudes in the vicinities of 58.50° W. Unlike the $r_P$, which correlates the data pairs, the $r_S$ correlates the ranks of the data [46] (p. 55), and so it provides us with information regarding the existence of a monotonic increasing or decreasing function that relates the correlated data. Given that both radiosonde stations were located between 57.50° W and 60.00° W, the fact that the two correlation coefficients were significant at these two latitudes is not an unexpected outcome. Furthermore, both coefficients being positive at these two particular longitudes indicates that the BI and the RBI were coupled in the sense that they increased or decreased in a concurrent fashion. This coupling faded away to the east to the extent that the correlations changed from positive to negative at a longitude in the 52.50° W–50.00° W band (the South Atlantic). To the

west, the longitude at which there was a change of sign in both coefficients was located in the 67.50° W–65.00° W band (the Argentine Patagonia shore). These easternmost and westernmost boundaries, determined by the correlations, can be interpreted as an average spatial extension of the blockings in the area.

**Table 4.** Pearson ($r_P$) and Spearman ($r_S$) correlation coefficients between the RBI at 58.50° W and the BI at different longitudes. The number of correlated pairs ($n$) is also shown. Values marked with an asterisk are significant to a 95% confidence level.

| Longitude | Correlation Coefficient | | $n$ |
| --- | --- | --- | --- |
| | $r_P$ | $r_S$ | |
| **67.50° W** | −0.28 | −0.53 | 9 |
| **65.00° W** | 0.09 | −0.28 | 11 |
| **62.50° W** | 0.46 | 0.41 | 15 |
| **60.00° W** | 0.69 * | 0.77 * | 16 |
| **57.50° W** | 0.61 * | 0.60 * | 18 |
| **55.00° W** | 0.36 | 0.47 * | 23 |
| **52.50° W** | 0.11 | 0.05 | 16 |
| **50.00° W** | −0.37 | −0.50 | 12 |
| **47.50° W** | −0.72 * | −0.77 * | 10 |
| **45.00° W** | −0.74 * | −0.78 * | 9 |

## 4. Discussion and Concluding Remarks

An updated climatology for the occurrence of blockings in the SH was conducted for a 74-year period spanning from 1974 to 2021. This was done by using the blocking criteria defined in [14], applied to daily 500 hPa geopotential height reanalysis data. Annual and seasonal distributions of blockings across the SH were organized in bands at every 10° of longitude. Irrespective of the duration of the events, three regions in which blockings were more prone to occur were identified. The primary region was located in the vicinities of the DL and had the maximum annual and seasonal frequencies of occurrences. The strongest events also took place there. In this area, the maximum frequency values varied from approximately 10% to roughly 14% of the total number of cases; the longitude at which this maximum took place was dependent upon the season. Actually, the mode of the distributions in Figure 5b–e had their easternmost shift in JJA. On an annual average, the absolute maximum was located at 160° W. Secondary-rank maxima were located in SSA and their surrounding waters. Blocking conditions there were found all year long, but there was an increase with respect to the annual average in MAM and in SON. Third-rank maxima were located east of the GM. The individual dates in which at least one blocking event had been located had a predominance of zonal wave 1 at both 35° S and 50° S on an annual average and were in DJF and JJA in most of the cases. These results are in concordance with the ones presented in [44]. However, it is worth noting, from Table 1, that shorter waves played a fundamental role in the remaining cases. The case study introduced in Figure 3 stressed that, aside from wave 1, the contribution of waves 6 and 7 were also important. The monthly distribution of episodes, irrespective of the longitude, showed that blockings in the SH took place all year long with maxima in JJA, which is in agreement with the existing literature [1]. Monthly distributions were also presented for every 20° of longitude. They revealed that 180° E had the most homogeneous distribution of episodes across the year.

The duration was also taken into consideration. The classification was carried out using "days of persistence" and by season for every 10° of longitude. The longest events (≥5 days) tended to be more concentrated in the surroundings of the DL. In order to better summarize the results, three different areas were defined according to [14]: PAC (110° E–80° W), ATL (70° W–0° W), and IND (10° E–100° E). In addition to presenting the annual figures for these areas, the seasonal figures (not discussed in [14]) were also introduced. Single-day events were by far more frequent in the PAC, both annually and seasonally, followed by the ATL and the IND. On the other hand, the longest located episodes

(11 and 12 days) were exclusively present in the PAC in JJA. It is worth mentioning that the maximum persistence is sensible to the blocking criteria. For example, the maximum duration was extended to 26 days in [30] by permitting the BI not to be strictly negative.

Linear trends were estimated for the annual and the seasonal time series at every 10° of longitude, both for the average BI and for the total number of events. Downward trends (i.e., stronger events) were detected in the annual BI time series at three different longitudes, two of which were in the PAC and the other in southwestern SA (ATL). Trends in the seasonal BI time series were found in the PAC only; all of them had a negative slope, with exceptions in DJF. Trends, in the annual time series, for the number of episodes were located in the PAC only with an overall positive slope, in discordance with the general downward trend found in [27] but using different blocking criteria. Downward trends in the seasonal time series, except in JJA, were found at points in the PAC. Particularly at 180° E, if the results shown in Table 3 are extrapolated, an increase in both the frequency (at a rate of 33 events century$^{-1}$) and in the strength of the blockings is expected in the future. In the ATL, there was an upward trend west of the GM in JJA, and mixed results were found further west in SON. No trends at all were located in the IND. Trends in the seasonal series were not evaluated in [27]. As of 2013, there was a medium level of confidence that the frequency of blockings from the model simulations would not increase in the SH, whereas the trends in their intensity remained uncertain [45] (p. 1248). Moreover, an apparent downward trend in blocking activity in the SH seemed to be consistent with an increasing trend in the Southern Annular Mode ([45] and references therein). Notwithstanding, increases might occur in certain regions [51] (p. 1003). It was further assessed in 2021 that the frequency of blockings in the Pacific sector was expected to decrease in DJF and SON [52] (p. 607). The results in Table 3 are in agreement with these findings.

It may be argued that the reanalysis dataset already assimilated radiosonde information along with other types of data, e.g., aircraft and satellite data [53]. In the construction of the alternative RBI, we attempted to use solely radiosonde information so that these data were isolated from the rest of the data that had been incorporated from the reanalysis database. This was quite difficult across the SH, given the lack of pairs of upper-air stations located at the appropriate coordinates in order to build a helpful circulation index that could resemble the BI. Given that SSA was the secondary region for blockings in the SH and that Ezeiza (34.81° S, 58.53° W) and the Falklands (Malvinas) (51.81° S, 58.45° W) met the criteria, an RBI was constructed in order to study the phenomena along 58.50° W and the surrounding areas. For this purpose, 500 hPa geopotential height data from these two upper-air stations, which encompassed the 1988–2021 period, were used so that the BI and the RBI could be compared to each other. Unfortunately, only 74 individual cases in which both indices were simultaneously negative could be paired, but nevertheless, we had enough information to gather some conclusions regarding the annual evolution of the RBI. At 60° W, 57.50° W, and 55° W, the RBI and the BI evolved in a concurrent fashion with a statistical significance, so they were related through an undetermined monotonic increasing function. Stronger relationships were found at 47.50° W and 45° W but with the function being monotonic and decreasing. These opposite relationships may be linked to areas of genesis and lysis. This is something that can be further investigated, for example, by cross correlating the BIs at different longitudes. Since this particular topic is beyond the scope of the present work it is proposed as a subject for future investigations.

The following points summarize our main findings:

- Blockings in the SH were all-year phenomena, with maxima in JJA;
- The surroundings of the DL were the primary area for their occurrence both annually and seasonally; the longest and strongest events also took place there;
- Second- and third-rank maxima were located in SSA and in the east of the GM; blockings were more prone to occur in SSA during MAM and SON;
- If the SH was divided into three different regions, the overall longest events occurred in the PAC during JJA;
- Wave 1 had the highest contribution, at both 35° S and 50° S, in most of the cases;

- Detected trends in the intensity were mostly negative (i.e., stronger events), and trends in the frequency of occurrence were upward in the western Pacific and in the eastern Atlantic and downward in the central Pacific and in the western Atlantic; all these trends are season-dependent;
- The RBI proved to be suitable for the description of blockings in the SSA area.

**Supplementary Materials:** The following supporting information can be downloaded at: https://www.mdpi.com/article/10.3390/atmos13091343/s1, Figure S1: Contribution of wave 2 in Equation (1) to the total variance of the 500 hPa geopotential height across the 74-year period in terms of seasonal means: (**a**) DJF; (**b**) MAM; (**c**) JJA; (**d**) SON; Figure S3: As in Figure S2 but for wave 3; Figure S4: As in Figure S2 but for wave 5; Figure S5: As in Figure S2 but for wave 6; Figure S6: Frequency distribution of blocking events persistent for 1 day (top), 2–4 days (middle) and ≥5 days (bottom) for every 10° of longitude: (**a–c**) annual; (**d–f**) DJF; (**g–i**) MAM; (**j–l**) JJA; (**m–o**) SON; infigure the season panels the corresponding annual frequencies are presented in grey bars for comparison purposes; the logarithmic vertical scale is homogenized throughout the panels.

**Author Contributions:** Conceptualization, A.E.Y.; methodology, A.E.Y.; software, A.E.Y.; validation, A.E.Y.; formal analysis, A.E.Y., S.G.L. and P.O.C.; investigation, A.E.Y.; resources, A.E.Y., S.G.L. and P.O.C.; data curation, A.E.Y.; writing—original draft preparation, A.E.Y.; writing—review and editing, A.E.Y., S.G.L. and P.O.C.; visualization, A.E.Y.; project administration, P.O.C.; funding acquisition, A.E.Y., S.G.L. and P.O.C. All authors have read and agreed to the published version of the manuscript.

**Funding:** This research was funded by Universidad Tecnológica Nacional, grant numbers PID MSUTNBA0006539 and PID MSINNBA0006543.

**Data Availability Statement:** The origin of the data used in this research is linked in the corresponding sections.

**Acknowledgments:** The helpful comments and suggestions made by three anonymous reviewers are much appreciated.

**Conflicts of Interest:** The authors declare no conflict of interest. The funders had no role in the design of the study; in the collection, analyses, or interpretation of data; in the writing of the manuscript, or in the decision to publish the results.

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
