# Peer review of "The Southern Hemisphere Blocking Index Revisited"

_atmosphere, doi:10.3390/atmos13091343_

Round 1

Reviewer 1 Report

This paper provides an overview of updated blocking results in the Southern Hemisphere using traditional assessment methods. Since blocking processes are important in terms of the formation of anomalous weather conditions, the updated blocking climatology can be used in studies related to the analysis of climatic extremes. The article is written in a good style, using the necessary terminology and a sufficient selection of references.

However, in order to improve the content and understanding of the article by readers, the following remarks should be made.

1. Does it make sense to analyze the contribution of waves to the variability of the H-500 at all latitudes of SH, based on the task of determining blocking at latitudes of 35-50 S (Fig. 2)?

2. Figure 3 should be supplemented with information on the time course of the BI (as a separate graph or diagram), so that the reader can track its changes in the study region.

3. Tables 2-3 are not necessary, a description in the text is sufficient.

4. The analysis of events up to 5 days (especially 1-2 days) in section 3.2 does not make sense, because it does not involve blocking according to the generally criterion that is indicated in your article. Or indicate in the text the importance of short-term periods (1-4 days) in the analysis of the overall dynamics of blocking, from your point of view.

5. The RBI parameter described in section 3.4 can hardly be called a blocking index, since condition (3) is not checked and criteria for blocking duration are not established. By itself, the appearance of a negative RBI cannot be indicative of a blocking process, although a subjective test of synoptic processes could give an answer as to how is really representative RBI<0. It would be desirable to illustrate this section by synotic maps with typical synoptic processes, indicating the position of sounding stations, because this information is interesting of practical value for operational work.

6. It is not clear on what basis the conclusion is made about the increase in the intensity and frequency of blocking in the future (line 603)? Only on the basis of a positive trend in the analyzed period?

Reviewer 2 Report

Review of «The Southern Hemisphere blocking index revisited» by Adrián E. Yuchechen et al.

Paper presents an updated climatology of the blocking characteristics in the Southern Hemisphere using NCEP/NCAR reanalysis for the 1948-2021. Blocking detection algorithm uses a daily 500-hPa data to create BI index. Additionally to reanalysis data, alternative blocking index were created using radiosonde data. This paper apposite to Atmosphere in terms of its topic and size.

In general, I have questions to data which author uses in the manuscript. Why do authors choose this reanalysis? I suggest to use state-of-the-art reanalyses (ERA5, CFSR, etc) from 1979 (satellite era) and compare blocking characteristics between these reanalyses. That would be interesting to know how well the reanalyses reproduces blockings in the SH.

I suppose the authors can cope with it quickly and they do not need to resubmit this article as a new regular paper.

Reviewer 3 Report

The authors use 74 years of reanalysis data to study distribution and trends of blocking events in the southern hemisphere. A blocking event is defined by a negative difference of the geopotential height (in m) at the two latitudes 35°S and 50°S, at the same (or similar) longitude. The authors further develop the geopotential field in Fourier series in order to see which wave patterns contribute to the occurrence of blockings.

The contents of the paper and the methods applied are surely correct, but to read the paper is tiring because of the many many single values that are presented for different seasons and different longitude zones. It is not possible to keep an overview, that is, after reading you have no coherent picture in mind. I admit, it is certainly difficult to describe the results in a fashion such that an overall imagination develops, but it is worth trying. In the current version, nothing remains after reading except the feeling that the distribution and trends of blockings is somehow chaotic. Therefore I suggest the authors try to make a very short summary with a few bullet points, listing their main findings (and leaving out any details).

Some specific comments and questions:

Section 3.1.1: How is the skewness defined here? If it is the usual definition from probability, what is your mean value? It would help to see the mathematical formula that you use.

Figure 4 (and similar): You mention 3rd rank maxima that can hardly be identified on the histograms. Please try whether using half-logarithmic plots can be used; they would show the higher rank maxima better.

Lines 284/285: Please use BI for the blocking index to be consistent with the rest of the text.

Line 291: You mention 4 longitude zones but list only three.

Section 3.1.2, line 346: A similar problem as before: Please define the gamma with a mathematical formula. I assume that this gamma is different from the gamma in 3.1.1, since here you consider monthly distributions. What is then the mean value, that is required as the basis of the 3rd central moment?

Section 3.3: I am surprised that you have so many missing data in a 74 years daily reanalysis time series. I think the reanalysis has no missing data. So my question is, what causes and why are the missing data?

Section 3.3 as a whole: It would be interesting to know what you think of these mixed results, upward here, downward there. What is your interpretation? Is it possible that the statistics cheats us, which means that there is physically nothing going on worth to mention although the trends are significant? Do we have a contradiction of statistical and physical significance here?

Line 534: I do not understand what you mean with "without making seasonal distictions"; I assume that you compare dates where the RBI is negative. What has this then to do with seasons?

Table 6: Again I am not completely aware of why you have such a low number of events? Is this because negative RBI and negative BI occur rarely on the same day? If so, can you then state a positive correlation at all?

Several occasions in the text: You sometimes write "mass of the distribution". I think what you mean is the maximum, which is usually called the "mode of a distribution" in probability theory.

Line 573: I am not sure whether "GM" was defined before. Please check!

Round 2

Reviewer 1 Report

Since tables 2-3 have been removed from the text, subsequent tables need to be renumbered.

Author Response

Thank you very much for the observation. The numbering of the tables is now correlative. The changes are highlighted in green.